# Sleepless in Gaza: War-related trauma and the neurobiological toll on sleep

Bilal Hamamra  and Fayez Mahamid 

An-Najah National University, Nablus, Palestine

## Research Article

war-related trauma; sleep disturbances; PTSD; children; qualitative research; Gaza strip

**Corresponding author:**
Fayez Mahamid;
Emails: mahamid@najah.edu

## Abstract

The Israeli war against Gaza has severely disrupted daily life, including sleep, a fundamental human need. Chronic war-related trauma has caused hyperarousal, nightmares, and insomnia, perpetuating psychological distress. Overcrowded shelters and limited mental health services exacerbate these challenges. This study examines how the Israeli war against the Gazans affected Gazans' sleep quality and patterns, focusing on sleep-related challenges faced by children and adults through firsthand accounts of war-induced trauma and stress. Forty semi-structured interviews with 20 children (ages 6–12) and 20 adults (14 mothers, 6 fathers) were analyzed using thematic analysis with a bottom-up, data-driven approach, refined through team discussions and cross-validation by independent judges. The five key themes identified are (1) chronic hypervigilance and sleep disruption, (2) trauma-driven sleep dysregulation in Gaza's children, (3) sleeplessness in shelters, (4) maternal vigilance and the ramifications of sleeplessness, and (5) the health toll of chronic sleep deprivation. The findings highlight the urgent need for culturally sensitive mental health interventions, improved living conditions, and family-centered support services to alleviate war-related insomnia in Gaza.

## Impact statements

The Israeli war on Gaza has severely disrupted residents' mental health and daily life, with sleep disturbances emerging as a major but often neglected consequence of conflict-related trauma. Findings from this study revealed how chronic hyperarousal, nightmares, and insomnia affect both children and adults, worsened by crowded shelters and limited access to mental health services. By capturing Gazans' lived experiences, the study underscores the need for trauma-informed, culturally sensitive interventions, improved living conditions, and family-centered support to reduce the psychological and neurobiological impacts of war-related sleep problems. Addressing these issues is crucial for individual well-being and the broader resilience of communities enduring prolonged conflict.

## Introduction

Exposure to violence and continuous trauma disrupts the sanctity of sleep, thrusting survivors into a cycle of insomnia, persistent nightmares, and incessant hypervigilance (Babson and Feldner, 2010; Ahmadi et al., 2022). The simple act of shutting one's eyes becomes a confrontation with lingering memories, where every night is a fight against the phantoms of sadness and grief (Harb et al., 2012; Pigeon et al., 2013). The continual rise of stress hormones disrupts the natural cycle of rest, guaranteeing that sleep is constantly disturbed by harsh recollections of fear (Slavish et al., 2022; Sheaves et al., 2023). In this trial of anguish, the mind is overwhelmed by an unyielding surge of cortisol and norepinephrine, ensnaring it in a harmful loop that exacerbates the scars of PTSD (Miller et al., 2017). The irony is striking: a place that used to be a refuge transforms into a platform for the deepest sorrows, with each restless moment resonating with the trauma's enduring impact (Babson and Feldner, 2010; Ahmadi et al., 2022).

Neurophysiological research indicates that trauma creates a lasting impact on the body's stress response systems. The activation of the HPA axis releases a surge of cortisol that disturbs the fragile structure of sleep (Babson and Feldner, 2010; Ahmadi et al., 2022), while heightened norepinephrine functions as a toxic flow that maintains the mind in an ongoing state of hyperarousal (Harb et al., 2012; Pigeon et al., 2013). This biochemical attack deprives the brain of its capacity to achieve restorative periods, trapping the sleeper in disrupted sleep and continuous discomfort (Germain et al., 2008; Miller et al., 2017). This neuroendocrine turmoil is a harsh reminder that trauma is not limited to distant recollections but persists in every uneasy moment, undermining any effort for tranquility (Babson and Feldner, 2010; Ahmadi et al., 2022).

Nightmares arise as the tormenting ghosts of trauma, ceaselessly showcasing scenes of unimaginable terror that cannot disappear into oblivion (Slavish et al., 2022; Sheaves et al., 2023).

Every nighttime vision serves as a harsh replay – a mirror displaying the deepest injuries of a damaged spirit (Phelps et al., 2008; Harb et al., 2012). These unwanted images compel the mind to experience horrors that destroy any hint of peace, transforming slumber into an interminable ordeal in which each subconscious second is besieged by the fear of former events (Phelps et al., 2008; Miller et al., 2017). In this anguished condition, the dream landscape shifts from a possible refuge into a conflict zone, where the specters of pain clash with the desire for recovery.

A range of measures has arisen to combat this unending assault on sleep. Cognitive-behavioral therapy for insomnia (CBT-I) stands out as a symbol of hope, reinstating the natural rhythm of sleep (Lancel et al., 2021; Williamson et al., 2021), whereas imagery rehearsal therapy (IRT) boldly transforms the disturbing tales of nightmares, easing their suffocating hold (Lancel et al., 2021). Pharmacological therapies – especially SSRIs – aim to calm the neurochemical storm that disrupts the mind (Germain, 2013; Williamson et al., 2021). When paired with exposure-focused therapies, like script-driven imagery that allows for guided confrontations with traumatic memories (Craske et al., 2008), these integrative approaches create a multi-faceted strategy against trauma's impact, providing a lifeline for those eager to restore restful sleep.

The effects of interrupted sleep are most evident in the Gaza Strip – a region marked by years of relentless conflict and political turmoil (Morina et al., 2018). In Gaza, the sounds of military strikes and explosions fill the night, transforming rest into a horror marked by relentless anxiety and deep hopelessness (Said and Hamid, 2017; El-Hani and Saleh, 2018). The ongoing weight of conflict raises stress hormones to harmful levels, disrupting the delicate equilibrium of rest and intensifying stress symptoms. In addition, the breakdown of community and family ties leaves survivors isolated, depriving them of the social comfort that could alleviate their pain (Hynie, 2018; Roberts et al., 2019).

The neurobiological impact of Gaza's unending conflict is clearly apparent. Persistent activation of the HPA axis floods the body with harmful amounts of cortisol, disrupting restorative sleep (Babson and Feldner, 2010; Ahmadi et al., 2022), while an excessively active amygdala sustains a continual state of fear (Harb et al., 2012; Miller et al., 2017). The compounded allostatic load further undermines the fragile framework of sleep, trapping survivors in an infinite loop of physical and emotional distress (Mesa-Vieira et al., 2022). In this bleak environment, the immune response is weakened by persistent inflammation – a consequence of relentless stress that reduces the capacity for recovery (Coalson, 1995; Diab et al., 2022; Chudzicka-Czupała et al., 2023).

Amidst this chaos, social media serves as a soothing sanctuary for the survivors of Gaza. Digital platforms change solitary suffering into a shared plea for assistance, with each post and tweet reflecting the anguish of restless nights and troubling dreams (Amro, 2024). A mother captures the anxiety: "I'm going to bed uncertain if I will make it through the night." These words obscure the distinctions between individual pain and collective defiance, elevating voices that seek worldwide acknowledgment and action.

Social support, essential for recovery, is weakened in Gaza. Compulsory relocation and ongoing strife have broken down conventional support systems, resulting in survivors being alone in their anguish (Hynie, 2018; Yildirim et al., 2020). However, programs that restore these ties through psychoeducation, peer assistance, and culturally sensitive mental health services provide a ray of hope within the gloom (Al-Krenawi and Graham, 2000; Yildirim et al., 2020). Joint initiatives between community leaders and mental health experts aim to establish a support network, forming safe spaces where those in distress can express their suffering and embark on the gradual journey of recovery (Hynie, 2018; Roberts et al., 2019).

### The settings

This study was conducted in the Gaza Strip during the Israeli war in the Gaza Strip in 2023–2024. Participants were internally displaced families living in overcrowded shelters, makeshift tents, and temporary housing under conditions of constant bombardment, insecurity, and scarcity of basic sleeping arrangements. These settings, characterized by unsafe and disrupted sleep environments, provided the context in which narratives of trauma and sleep disturbances were explored. Considering prior studies showing that Palestinians in the Gaza Strip encounter various types of war-related trauma, political violence, and sleep issues due to the ongoing conflict, this research aimed to examine how these conditions affected sleep quality and patterns among displaced families. Specifically, the study focused on the lived experiences of women and children, investigating: (1) *What are the effects of the recent conflicts in Gaza and associated distressing incidents on the sleep quality and patterns of local inhabitants?* (2) *In what ways do Palestinian refugees in Gaza describe their sleep experiences, and what major obstacles or issues do they face in obtaining restorative sleep?*

### Methods

### Participants

This study employed a qualitative approach using grounded theory (Corbin and Strauss, 2015), which seeks to uncover the beliefs and meanings underlying actions, evaluate both rational and irrational behaviors, and demonstrate how logic and emotion interact to shape individuals' responses to events and problem management. Participants in this study are 40 native Arabic-speaking Gazans, including 20 children aged 6–12 years (mean age = 8.4, SD = 2.3) and 20 adults – 14 mothers (mean age = 33.2, SD = 4.5) and six fathers (mean age = 36.4, SD = 5.1). These participants provided critical insights into the psychological toll of ongoing violence. Their narratives detailed traumatic experiences of fear, loss, and relentless disruption of daily life, underscoring the severe mental health consequences of continuous exposure to conflict. Qualitative data were gathered via 40 semi-structured interviews held in internally displaced camps in Rafah, particularly in shelter schools during the war. Local research assistants, who experienced displacement and security threats themselves, served as gatekeepers and utilized a snowball sampling approach to gather participants. They received a thorough briefing on the study's objectives, ethical guidelines, and interview methods, ensuring that the survey questions aimed to reduce emotional discomfort. The study received approval from the An-Najah Institutional Review Board (IRB) prior to data collection. All participants, including children, provided informed consent or assent, with parental consent obtained for minors. Participants were fully informed about the study's purpose, procedures, potential risks, and their right to withdraw at any time without penalty. They were made aware of their right to withdraw at any moment, and mental health professionals were available to provide prompt assistance, with contact information for local support services provided for post-interview care. Despite considerable logistical challenges, including sporadic power failures, rapid evacuations, airstrikes, and communication disruptions, the research team adhered to strict ethical protocols and employed flexible data-gathering methods, ensuring both the reliability of the findings and the well-being of

participants. Confidentiality and anonymity were strictly maintained, and all data were securely stored. Given the sensitive nature of the study, particularly regarding trauma and sleep disturbances, all precautions were taken to minimize psychological distress.

## Instruments and procedures

The qualitative data were gathered from 40 semi-structured interviews with Palestinian refugees living in displacement camps in Rafah amid the recent conflict in the Gaza Strip. Both the interviewer and all participants were fluent in Arabic as their native language. In each camp, local research assistants served as gatekeepers, enlisting participants through a snowball sampling method. At the outset, research assistants were informed about the study's objectives, the overall count of interviews planned, and the process for recruitment. The interview questions were thoughtfully written to reduce emotional discomfort, and participants were advised they could stop their involvement at any point if they felt uneasy. A qualified mental health professional was available to provide immediate support to any participant experiencing distress, and information about local mental health services was offered for follow-up care if needed. Every interview, concentrating on trauma from war and sleep issues in refugees, lasted between 35 and 60 min, with the majority lasting about 50 min.

## Data analysis

All interviews were audio recorded and transcribed verbatim into Arabic by a native-speaking researcher. The transcripts were then analyzed using thematic content analysis (TCA) (Braun and Clarke, 2014) to identify the primary themes emerging from the data. A bottom-up, data-driven approach was applied to extract categories from the raw text (Bradley and James, 2020). The process involved open coding to derive initial themes, subsequent organization of these themes into a structured framework, and final refinement through discussions with five independent judges to ensure consensus on the categories and subcodes.

## Trustworthiness and credibility

To ensure the credibility and rigor of the study, a multi-step strategy was implemented. First, a total of 40 semi-structured interviews with children and adults provided rich, diverse data. Triangulation across participant groups and team discussions enhanced the depth and validity of findings. Member checking allowed participants to verify preliminary interpretations, while peer debriefing and independent cross-validation by external judges reduced potential bias. An audit trail documented all stages of data collection, coding, and theme development, ensuring dependability. Throughout the process, the lead researcher engaged in reflexivity, continually reflecting on personal assumptions and positionality. Finally, coding reliability was assessed, yielding 92% consistency with the original coding and a Cohen's kappa of 0.94, demonstrating that the thematic analysis was both reliable and replicable. Together, these steps form an interconnected framework that strengthens the overall trustworthiness, transparency, and rigor of the study.

## Results

Thematic content analysis of the interview transcripts led to the identification of five main themes: (1) *chronic hypervigilance and sleep disruption*, (2) *trauma-driven sleep dysregulation in Gaza's children*, (3) *sleeplessness in shelters*, (4) *maternal vigilance and the ramifications of sleeplessness, and* (5) *the health toll of chronic sleep deprivation*.

## Theme one: Chronic hypervigilance and sleep disruption

In Gaza, nights have become a relentless ordeal where sleep is elusive and overshadowed by a persistent fear of death. One mother expressed her inner turmoil, stating, *"I'm grateful to be alive, but I am completely shattered mentally."* A young girl captured the pervasive terror by saying, *"Every time I hear an explosion, I feel like I'm going to be killed just like my friend Dima."* Another child recounted, *"I didn't sleep all night because of nightmares; here in Gaza, I keep running from place to place to escape the bombing."* One more young voice added, *"I sleep dreaming of the faces of the children I see in videos, along with corpses and destruction."* Adults, too, are not spared; one man, 34, confessed, *"I still wake up at 3 AM every night, haunted by the echoes of bombs. The constant barrage of explosions makes every moment a fight for survival."* A 40-year-old man shared, *"After every siren, my heart races and I'm left with an overwhelming sense of dread. The trauma has seeped into every part of my life, leaving me perpetually anxious."* A father said that *"With bombs falling every night, the trauma deepens, and children – even in their sleep – wish for death to end the nightmare."* One mother lamented, *"Each time I close my eyes, I think of the bombs falling. Sleep is no longer sleep; it's a step closer to death."* Another mother tried to comfort her children by saying, *"I can't promise them safety, but I stay awake beside them, praying they'll make it through the night."* The constant fear is palpable, as every whispered sound in the dark reinforces the dread that transforms each moment of rest into a battleground for survival. This collective testimony illustrates a community where fear and uncertainty have rendered sleep a scarce and dangerous refuge.

## Theme two: Trauma-driven sleep dysregulation in Gaza's children

For Gazan children, sleep has become a stage for nightmares that mirror the brutal realities of their everyday lives. One mother tearfully stated, *"My children are afraid to sleep, and I am afraid for their lives too."* Another grieving mother, having lost several of her children to bombings, revealed, *"Every time I close my eyes, I see my children in front of me, so I'm afraid to sleep."* A mother recounted, *"I hear my daughter cry in her sleep, calling out for her father,"* while a father described his son's agony: *"He wakes up screaming, asking if we're going to be hit tonight. I don't know what to say. There is no comfort."*

A ten-year-old girl confessed, *"My dreams have changed – they used to be better. Now, every dream fills me with fear, and I wake up unable to sleep."* Another child remarked on the loss of a secure home environment, stating, *"Losing one's parents means the loss of security and love,"* further compounding their inability to find solace in sleep. Every night, these children relive traumatic events in their dreams.

The shattered sleep of Gaza's children starkly indicates the overwhelming trauma that pervades their lives. One child stated, *"The children in Gaza are deprived of their childhood due to the lack of security and stability in their surroundings."* A 12-year-old girl shared, *"I feel frustrated, and nightmares constantly haunt me about the day when soldiers killed my parents and several of my relatives."* Another child, just 6 years old, said, *"I barely say a word except to*

ask about my mother and my four siblings who were killed in the airstrikes." Her father reflected, "She used to be full of life, but now she keeps asking about her mother. We tell her, 'Mama is in heaven – don't play and don't scream when someone approaches you."

A father observed, "Many children in Gaza wet their beds due to trauma, which indicates the depth of their psychological wounds." A mother expressed despair, "We once hoped for a bright future for our children, but the wars have turned their dreams into nightmares." Another mother lamented, "The war has stolen our children's laughter, and now every explosion brings back bitter memories."

### Theme three: Sleeplessness in shelters

The disruption of sleep in Gaza is not solely a product of external violence; it is also deeply influenced by the dire living conditions in overcrowded shelters. One father described the harsh reality: "The situation in the schools is horrible. They are overcrowded, with no toilets, no food, no water, and no privacy whatsoever. So I decided to come back with my family to my bombed house and live in whatever space was left standing." Another mother explained the logistical challenges they face: "We sleep in shifts now, because we can't all close our eyes at the same time. Who will watch over us?" Another father summed up the ever-present alertness with, "I'm sleeping with one eye open," a phrase that encapsulates the constant state of vigilance required in these environments. The lack of privacy and basic amenities means that even if the threat of explosions were momentarily absent, the physical discomfort and insecurity persist, preventing restorative sleep. These testimonies underscore that the trauma of conflict is compounded by the everyday realities of life in overcrowded, unsafe shelters. The relentless noise, the cramped conditions, and the ever-present reminder of displacement transform every attempt at sleep into a struggle against both physical discomfort and psychological distress.

### Theme four: Maternal vigilance and the ramifications of sleeplessness

Gazan mothers have emerged as tireless guardians, sacrificing their own rest to protect their children. One mother shared, "When children tell me about the sound of the shelling, I remember how our home shook during the explosions. I share their anxiety and fear." Another expressed her burden, saying, "I have not received any psychological support, even though I now try to help these children." A Gaza father encapsulated the phenomenon of maternal vigilance by observing, "In Gaza, a mother does not sleep; she remains in constant alertness, listening to the darkness, inspecting its margins, and discerning each sound to craft a lullaby of reassurance for her children." Such powerful imagery reflects the enormous responsibility these women shoulder. They remain awake long after their children have fallen asleep, their eyes scanning the dark for any sign of danger. Their sleepless nights are a sacrifice made in the hope of providing even a modicum of safety, even as the shadows of conflict loom large. The maternal role in Gaza has evolved into one of both nurturer and vigilant protector, an unyielding force against the encroaching despair that threatens to rob their children of any semblance of a peaceful future.

### Theme five: The health toll of chronic sleep deprivation

The relentless disruption of sleep is taking a heavy toll on the physical and mental health of Gaza's residents. One mother stated, "My body is giving up. I feel tired all the time, but I can't rest; sleep has become a foreign thing." Another mother described her decline, remarking, "I feel my body weakening, and my mind is foggy; it's like I'm slowly fading." A father noted, "I can recognize the bombs by their sound now. We no longer sleep; we wait." These statements highlight the severe physical exhaustion that results from chronic sleep deprivation. A father said that "The relentless nightly bombardment crushes any hope of safety – leaving families to face each night as if it were their last." A mother said that "There is a noticeable deterioration in the psychological condition of children in Gaza." The dire living conditions and continuous exposure to trauma mean that even basic restorative sleep is unattainable, leaving individuals in a perpetual state of physical and mental depletion. One mother said that "Many children in Gaza wet their beds, suffer from stuttering and nightmares, and refuse to eat. The question that haunts many of them now is: When will the next war come? What will we do? And where will we go?" This unending cycle of insomnia and trauma not only undermines physical health but also casts a long shadow over the future well-being of an entire generation.

## Discussion

The participants' intense narratives of restless nights filled with ongoing fear reflect the findings of Babson and Feldner (2010), who highlight that persistent trauma triggers the hypothalamic-pituitary-adrenal (HPA) axis, resulting in increased cortisol levels that disturb the sleep cycle. Viewed through PTSD-sleep reciprocity models, persistent nocturnal arousal plausibly amplifies daytime intrusions, irritability, and attentional deficits, establishing a bidirectional cycle that sustains insomnia under continuing threat (Babson and Feldner, 2010; Germain, 2013; Pigeon et al., 2013). Analytically, these narratives evidence a persistent nocturnal hyperarousal that extends beyond immediate bombardment, thereby extending physiological accounts by specifying conditioned night-time cues as maintaining factors. This biochemical imbalance reflects the accounts of Gazans, who portrayed sleep as a conflict zone overwhelmed by anxiety and heightened alertness. The continuous activation of stress response mechanisms, as outlined by Ahmadi et al. (2022), guarantees that the body's state of vigilance remains, rendering peaceful sleep unattainable – an experience reflected in the accounts of those tormented by the sounds of explosions and the constant fear of impending violence. This represents a wider recognition that the physiological effects of trauma greatly hinder the ability to achieve restorative rest. These accounts indicate a hyperarousal-based sleep phenotype in which downregulation remains impaired even during relative lulls, as nocturnal cues acquire conditioned threat salience that sustains sympathetic activation; accordingly, conflict-congruent care should prioritize physiology-forward components (e.g., paced breathing, low-stimulus pre-sleep routines, and contextually salient safety cues) alongside cognitive strategies while exposure to danger persists.

The accounts of children in Gaza, whose slumber is interrupted by persistent nightmares, correspond with Miller et al. (2017), who showed that initial exposure to violence interferes with neural pathways crucial for regulating emotions and consolidating memories. Conceptually, these dreams function simultaneously as neurobiological expressions of fear memory and as socio-moral narratives through which families articulate loss, responsibility, and survival (Coalson, 1995; Phelps et al., 2008; Harb et al., 2012; Miller et al., 2017). The experiences of night terrors among the young participants, filled with images of destruction and grief, highlight

this neurodevelopmental susceptibility. These results reflect the findings of Phelps et al. (2008), who observed that post-traumatic nightmares act as harsh reenactments of traumatic experiences, further hindering cognitive and emotional growth. This trend is reflected in Williamson et al. (2021), who highlight that caregivers in war-torn areas endure increased stress, forgoing their own rest to ensure a sense of safety for their families, thus worsening the cycle of trauma across generations. In this context, nightmares operate simultaneously as a rehearsal of fear memories and as familial testimony, which warrants adapting imagery-based interventions to incorporate culturally anchored meaning-making while embedding delivery within routines that protect caregiver sleep.

The effects of crowded shelters on sleep, as outlined by the participants, resonate with the findings of Slavish et al. (2022) and Sheaves et al. (2023), who reported how external stressors, like noise and reduced privacy, worsen internal physiological reactions. Analytically, the sleep ecology of displacement – noise, light, thermal discomfort, and crowding – interacts with hyperarousal to shorten cycles and increase awakenings independent of individual predispositions (Germain et al., 2008; Slavish et al., 2022; Sheaves et al., 2023). The accounts of residents regarding sleeping in intervals, constant alertness, and physical unease correspond with these studies, highlighting that sleep disruption caused by trauma is frequently exacerbated by severe living conditions. This convergence of environmental and neurobiological stressors highlights the complicated situation encountered by Gaza's displaced community, where even instances of possible relief are undermined by the constant reminders of war and displacement. These conditions reflect the findings of Germain et al. (2008), who emphasize that environmental instability worsens trauma symptoms and disrupts sleep. An environmental lens, therefore, explains fragmentation beyond individual pathology and justifies specifying sleep-conducive features – predictable quiet hours, light control or blackout materials, modest privacy partitions, ear/eye protection, and basic thermal regulation – as integral components of humanitarian health standards.

Mothers, as protectors during restless nights in Gaza, illustrate the dual challenges pointed out by Lancel et al. (2021), indicating that caregivers in areas of conflict face increased psychological stress that disturbs their sleep and affects family relationships. This nocturnal labor is unevenly distributed, yielding a gendered concentration of physiological burden that is both clinically salient and socially patterned (Jones and Abdullah, 2019; Roberts et al., 2019; Lancel et al., 2021). The maternal vigilance presented in the accounts reflects the conclusions of Miller et al. (2017), who observed that the intergenerational transmission of trauma is frequently influenced by disturbed sleep patterns in families. This maternal sacrifice emphasizes the results of Roberts et al. (2019), who pointed out that lacking sufficient social support worsens psychological distress, demonstrating the cyclical characteristics of trauma in families impacted by war. A family-systems perspective clarifies that protective co-sleeping and night watching reduce children's distress yet maintain adult hypervigilance, concentrating physiological costs on caregivers – particularly mothers – so programmatic measures should include rotation of night duties, brief protected off-duty intervals, child wind-down rituals, and mobilization of community support to mitigate cumulative sleep debt.

The health effects of long-term sleep deprivation noted by residents of Gaza resonate with the findings of Germain et al. (2008), which explained how extended sleep interruption increases allostatic load, resulting in cognitive decline, reduced immunity, and greater susceptibility to mental health issues. The observed

pattern is consonant with an allostatic-load account in which cumulative sleep loss mediates the association between chronic threat exposure and multisystem dysregulation (Germain et al., 2008; Mesa-Vieira et al., 2022; Chudzicka-Czupała et al., 2023). The participants' descriptions of physical fatigue, cognitive cloudiness, and declining health correspond with the idea of allostatic overload, as explained by Mesa-Vieira et al. (2022), where ongoing stress speeds up physiological decline. This reflects the findings of Chudzicka-Czupała et al. (2023), who reported comparable health effects in conflict-affected groups, highlighting the necessity for urgent psychological and medical responses. These outcomes constitute a plausible pathway through which violence degrades everyday functioning in learning, caregiving, and judgment; future inquiries should pair qualitative testimony with feasible physiological proxies (brief sleep diaries and low-burden wearables) to quantify recovery trajectories and evaluate low-intensity interventions.

The pressing demand for psychological assistance, highlighted by both mental health experts and community members, emphasizes the vital necessity of combining personal therapeutic measures with comprehensive socio-political changes to alleviate the diverse effects of war-induced trauma on sleep and general health. Operationalization entails codifying sleep-protective standards in shelters (e.g., quiet-hour governance, lumen limits and light control, safe child-sleep zones) and training lay providers to deliver brief, culturally adapted sleep-support protocols within community settings (Hynie, 2018; Roberts et al., 2019; Yildirim et al., 2020; Lancel et al., 2021). This is consistent with the findings of Al-Krenawi and Graham (2000), who support the need for culturally aware mental health services in areas of conflict. The accounts and writings emphasize that tackling sleep issues caused by trauma necessitates a comprehensive strategy, merging therapeutic methods with broader initiatives to elevate living standards and build community strength, thus improving the prospects for lasting psychological healing and wellness in Gaza. A two-level response is therefore indicated: task-shifted, brief clinical packages foregrounding environmental modification, somatic down-regulation, and child-focused bedtime routines, and structural measures that secure nighttime safety and dignified living conditions so that clinical gains are durable; conceptualizing "sleep protection" as an element of civilian protection provides a coherent bridge between health services and humanitarian design.

### Limitations

This study has several limitations that should be acknowledged. First, the use of convenience sampling among internally displaced families in Gaza restricts the transferability of the findings beyond similar war-affected contexts. This is a typical limitation of qualitative research, which emphasizes depth of understanding over generalizability. Second, the data relied on semi-structured interviews and self-reported accounts from both children and adults, which may be influenced by recall bias, social desirability, or children's limited ability to articulate complex experiences. Third, data collection took place during the height of the Israeli war on the Gaza Strip, a context that likely magnified sleep disturbances and psychological distress, thereby shaping participants' narratives. Future studies could strengthen rigor and trustworthiness by using multiple sources of data (e.g., sleep diaries, observational methods, or physiological measures) to triangulate self-reports. In addition, longitudinal qualitative designs would allow researchers to trace how war-related sleep challenges

evolve over time, providing a deeper understanding of both immediate and long-term consequences.

## Conclusion

This study elucidates the profound and multifaceted impact of continuous war-related trauma on sleep among Gazan residents. The thematic analysis revealed a distressing pattern of chronic hypervigilance, pervasive sleep dysregulation in both children and adults, and the exacerbating influence of overcrowded shelter conditions. Notably, the narratives of maternal vigilance underscore a dual burden: while mothers strive to protect their children, their own health deteriorates as a result of chronic sleep deprivation. Together, these findings reinforce the critical notion that the neurobiological toll of trauma – characterized by persistent HPA axis activation and resultant biochemical imbalances – directly compromises restorative sleep, thereby perpetuating a cycle of psychological and physical distress.

The study's insights highlight an urgent need for culturally sensitive mental health interventions that not only address the neurophysiological disruptions associated with PTSD but also mitigate the environmental stressors that undermine sleep quality. Future research should further examine the longitudinal effects of sustained sleep deprivation on cognitive and emotional development, particularly among children, while policy initiatives must prioritize the enhancement of living conditions in displacement settings. Ultimately, these integrated efforts are vital to fostering resilience and facilitating recovery among populations enduring the relentless hardships of conflict.

**Open peer review.** To view the open peer review materials for this article, please visit http://doi.org/10.1017/gmh.2026.10164.

**Data availability statement.** The datasets generated during and/or analyzed during the current study are available from the corresponding author on reasonable request.

**Author contribution.** The authors were responsible for all aspects of the research, including conceptualization, design, data collection, data analysis, and manuscript preparation. The authors have read and approved the final version of the manuscript.

**Financial support.** No funding was received for this study.

**Competing interest.** The authors declare no competing interests.

**Ethics approval and consent to participate.** All procedures performed in this study involving human participants were in accordance with the Institutional Review Board (IRB) of An-Najah National University, the American Psychological Association (APA, 2010), and the 2013 Helsinki Declaration.

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
