## [Reviewer Report]

thank you for such a novel important issue, yet there are some points need to be clarified for the rigour of the study

1-in the introduction problem statment and signficance of the study should be more apperant .add the major objective

in method section the study design justifies the need for such design, it should also begin with first then followed by the sampling method

2-Organize ethical considerations in one paragraph

3-in method section ensures credibility items, number of interviews, triangulation, member checking and peer debriefing, and auditing are missing, researcher reflexivity

4- it would be better to replace cost of sleeplessness (cost) with ramcification or conseque

---

## [Reviewer Report]

Thank you for writing such a nice and relevant article. I have a few important comments and suggestions that I believe should be addressed:

Study Limitations: The manuscript would benefit from a clearly defined section discussing the study’s limitations. This is essential for contextualizing the findings and acknowledging methodological constraints.

Informed Consent: you need to clarify how informed consent was obtained from participants, including whether it was written, verbal, or implied, and whether ethical approval was granted by a relevant board or institution.

APA Referencing: Please ensure that all in-text citations and references in the reference list adhere strictly to APA formatting. Several inconsistencies were noted.

Keywords: The keywords should be revised to better reflect the core themes of the study and to improve the manuscript’s indexing and discoverability.

Mental Health Support: you should indicate whether mental health or debriefing support was made available to participants who may have experienced distress as a result of the study, especially if it addressed potentially sensitive or traumatic topics.

Missing Relevant Literature: The manuscript does not cite important existing Palestinian research in this field. I strongly recommend including references to the work of Qouta, Diab, and Veronese, among others, whose contributions provide valuable context and depth to the topic.

---

## [Reviewer Report]

Review of: Sleepless in Gaza: War-Related Trauma and the Neurobiological Toll on Sleep

GMH-2025-0137

This paper makes a modest contribution by drawing attention to a particular aspect of the tragedy of Gaza, profound sleep problems. In addition to the call for psychological and medical services, I would add the obvious that such services will be, at best, palliative until the continuing community trauma is resolved.

I would invite the authors to add some discussion of how the sleep disorders in Gaza may be different, more exaggerated, etc. than those observed in non-traumatized settings, or how observing in these extreme settings illuminates perhaps our more general understanding of sleep disorders.

Understanding both a) the horrible events that have been imposed on Gaza and b) the sensitivities around and controversial nature of appraisals of those events and their responsibility, I would encourage the authors not to use the word “genocide” to describe the war in Gaza. In other ways, they might make the language more fact based and less judgmental or dramatic as with, e.g., “tormenting ghosts of trauma.”

The Discussion needs revision. As it stands, it reads like an extended annotated bibliography. It should be better organized to identify and develop key themes and then use references to explain or elaborate on those themes.

The references need to be checked throughout to assure accuracy. I did not check them all, but noted trouble with two in which I had interest:

First example: Reference as included in the text:

• Rogowska, A. M., & Pavlova, I. (2023). Fear, nightmares, and insomnia: PTSD and sleep disorders in Ukrainian students during the Russia‐Ukraine war. Sleep Medicine, 108, 68–76. https://doi.org/10.1016/j.sleep.2022.07.059

• Clicking on the url yields “DOI Not Found”

• Examining the Sleep Medicine portal on the web,https://www.sciencedirect.com/journal/sleep-medicine/vol/108/suppl/C i found no such article

• Googling the authors and article title yielded:

• Aleksandra M Rogowska 1, Iuliia Pavlova 2. Psychiatry Res. 2023 Oct:328:115431. doi: 10.1016/j.psychres.2023.115431. Epub 2023 Sep 7. A path model of associations between war-related exposure to trauma, nightmares, fear, insomnia, and posttraumatic stress among Ukrainian students during the Russian invasion. PMID: 37688837 DOI: 10.1016/j.psychres.2023.115431

• The actual article seems similar in title/scope to the reference

Second example: Reference as included in the text:

• Yildirim, S., Demir, N., & Ünlü,, G. (2020). The role of social support and resilience in the mental health of Syrian refugees: An exploratory study. Community Mental Health Journal, 56(3),341–349. https://doi.org/10.1007/s10597-019-00522-7

• Clicking on the url yields “DOI Not Found”

• I found no such article in the website of the Community Mental Health Journal.

• PubMed found no hits for: Yildirim [au] AND Demir [au] AND social support [tiab]

---

## [Editor Report]

In addition to the valuable points raised by the reviewers, I would kindly suggest considering a slight rephrasing of the first research question or clarifying the link between the question and the presented results. As it is currently formulated, the question “First: In what ways have the recent conflicts in Gaza and the associated distressing incidents affected the overall sleep patterns and quality for local inhabitants?” may give the impression of an investigation into changes before and after the onset of the conflict. This, however, does not seem to be the primary focus of the study.